# Thermodynamic driving forces in contact electrification between polymeric materials

Hang Zhang [1], Sankaran Sundaresan[2] & Michael A. Webb [2] ✉

Contact electrification, or contact charging, refers to the process of static charge accumulation after rubbing, or even simple touching, of two materials. Despite its relevance in static electricity, various natural phenomena, and numerous technologies, contact charging remains poorly understood. For insulating materials, even the species of charge carrier may be unknown, and the direction of charge-transfer lacks firm molecular-level explanation. Here, we use all-atom molecular dynamics simulations to investigate whether thermodynamics can explain contact charging between insulating polymers. Based on prior work suggesting that water-ions, such as hydronium and hydroxide ions, are potential charge carriers, we predict preferred directions of charge-transfer between polymer surfaces according to the free energy of water-ions within water droplets on such surfaces. Broad agreement between our predictions and experimental triboelectric series indicate that thermodynamically driven ion-transfer likely influences contact charging of polymers. Furthermore, simulation analyses reveal how specific interactions of water and water-ions proximate to the polymer-water interface explain observed trends. This study establishes relevance of thermodynamic driving forces in contact charging of insulators with new evidence informed by molecular-level interactions. These insights have direct implications for future mechanistic studies and applications of contact charging involving polymeric materials.

Contact electrification, or contact charging, is a widely observed phenomenon that results in static charges present on materials based on their touching[1-7]. In nature, such charging manifests in dust storms, which generate substantial charge via collisions of sand particles[8,9], and in ash plumes of volcanic eruptions, which accumulate and release charge in the form of volcanic lightning[10]. In modern technology, contact charging enables xerographic printing[11,12] and energy generation in wearable devices[13,14]. Undesirable charging also underlies issues in several industrial applications[15,16], such as wall-sheeting in reactors[17], disruption of particle mixing[18] and hazardous electrostatic discharge[19]. Despite this prevalence, precisely how and why contact charging occurs in many scenarios remains ambiguous. Therefore, understanding contact charging is of interest to advance fundamental science and to enhance technological processes[20-22].

The mechanism of contact charging strongly depends on the nature of the charge carriers, the materials, and the environment. Three modes of charging include electron transfer[6,23-25] wherein surface work functions direct charge transfer, ion transfer[3,26] wherein intrinsic or acquired mobile ions transfer between materials, and material transfer[27] wherein charged pieces of material physically move between surfaces. While electron transfer dominates charging of metals[2] and semiconductors with small band-gaps, the presence of insulating layers atop materials can obfuscate understanding predicated solely on work functions[7]. Moreover, contact charging of insulating materials themselves, such as polymers[28,29], likely requires other charge-carrier species. One compelling hypothesis is that unequal transfer of cations and anions between materials results in sustained, asymmetric charge accumulation on surfaces[3]. This mode

[1]Department of Chemistry, Princeton University, Princeton, NJ 08544, USA. [2]Department of Chemical and Biological Engineering, Princeton University, Princeton, NJ 08544, USA. ✉e-mail: mawebb@princeton.edu

requires that materials must either natively possess or otherwise acquire mobile ions, raising questions as to what ions are present.

Water-ions—hydronium ($H_3O^+$) and hydroxide ($OH^-$)—are viewed as potential charge-carriers underlying contact charging of insulating materials[3,30]. Water is almost ubiquitously present, in real-world and experimental systems alike, having been detected across diverse chemical surfaces and a broad range of conditions[31-38]. Mosaic patterns of charge on polymer surfaces following contact have been attributed to the presence of water patches[39], as water has been observed to only partially cover surfaces, forming patches or islands[37,38]. Effects of relative humidity on electrostatic charging highlight a potential role of water and its ions[28,30,38,40] as do numerous studies related charging phenomena directly at liquid-solid interfaces[41-45]. Furthermore, there are existing correlations between water-related properties and contact charging of polymers, such as acid/base dissociation constants[46], Lewis acidity or basicity of polymers[47], and zeta potentials of non-ionic polymers[3,48]. While such work establishes a potential role of water and associated ions in many circumstances, why water-ions should concentrate on a certain material after contact with another is unclear.

Various theoretical and conceptual frameworks have been constructed to explain water-ion transfer as a mechanism for contact charging of polymers. For example, a lattice model introduced by Grosjean et al.[49] quantitatively accounts for mesoscale spatial correlations that might explain contact charging between polymer surfaces of the same chemistry. Jaeger and coworkers examined the role of surface hydrophilicity on charging, finding consistency with models premised on $OH^-$ diffusion between adsorbed water patches with asymmetric coverage on the contacting surfaces[33,50]. Nevertheless, these models generally lack nanoscopic attributions to specific molecular-level underpinnings. Although molecular simulation techniques, such as density-functional theory and ab initio molecular dynamics, have been deployed to unravel complex nanoscale phenomena of contact charging in systems comprised of crystalline minerals, MXenes, oligomers, and water[26,51-54], studies involving polymers are nascent.

In this study, we employ molecular dynamics simulations to investigate whether thermodynamic driving forces for water-ion transfer can feasibly impact contact charging of insulating polymers. We hypothesize that polymer surfaces present distinct nanoenvironments for water molecules and water-ions that result in chemical-potential differences, which govern asymmetric transfer of ions between surfaces upon contact. To test this hypothesis, we utilize thermodynamic integration[55] to extract relative free energies of $H_3O^+$ and $OH^-$ on polymers of varying hydrophilicity[56]. These free energies, which are sensitive to polymer chemistry and underlying molecular interactions, provide a basis to predict the direction of ion-transfer between polymer surfaces. Such predictions enable construction of a triboelectric series based entirely on thermodynamic driving forces, which intriguingly illustrates good agreement with experimental triboelectric series. Further simulations that directly probe ion partitioning between two surfaces illustrate similar trends. This consistency establishes the viability of thermodynamically driven water-ion transfer in contact charging of polymers. Furthermore, the methodology highlights molecular-level nuances that may hold other implications for contact charging and general understanding of water-polymer interfacial interactions.

## Results

### Hypothesis of thermodynamically driven water-ion transfer

The possibility of contact charging as a process driven by the relative ion-surface affinities has been considered since at least the 1950s[57], although molecular evidence is scarce. Here, we consider whether the free energies of $H_3O^+$ and $OH^-$ within droplets on different polymer surfaces (Fig. 1A) are predictive of contact charging (Fig. 1B). The posited mechanism of charging is that (i) water droplets on surfaces

contain $H_3O^+$ and $OH^-$ with chemical potentials that depend principally on surface chemistry but also other factors (e.g., preexisting ion concentration, humidity, electric fields, etc.), (ii) water-ions can diffuse between surfaces when they are sufficiently close, and (iii) the relative abundance of water-ions on two surfaces following diffusion events is biased by the relative chemical potentials. Here, water ions may arise from ambient water, as suggested by previous experimental studies[58,59], but all calculations are agnostic to their precise origin.

Figure 1B illustrates contrasting scenarios of water droplets present on surfaces, $A$ (blue) and $B$ (red), that guide our calculations. In reference State I, droplets are neutral on both surfaces. In State II, contact yields a charge-separated pair where $H_3O^+$ resides on $A$ and $OH^-$ resides on $B$; the free energy of State II relative to State I is $F_{AB}^{+-}$. In State III, contact yields a charge-separated pair, which is the reverse of State II; the free energy of State III relative to State I is $F_{AB}^{-+}$. These free energies are obtained as $F_{AB}^{+-} = F_A^+ + F_B^-$ and $F_{AB}^{-+} = F_A^- + F_B^+$ where $F_S^\alpha$ indicates the free energy of adding an ion of type $\alpha \in [+,-]$ to surface $S \in [A,B]$ (Fig. 1B, bottom). The difference $\Delta F_{AB}^{+-} \equiv F_{AB}^{+-} - F_{AB}^{-+}$ reflects an initial thermodynamic driving force for contact charging. In particular, $\Delta F_{AB}^{+-} < 0$ indicates greater likelihood for surface $A$ to become positively charged and surface $B$ negative compared to the opposite, while $\Delta F_{AB}^{+-} > 0$ indicates greater likelihood for surface $A$ to become negatively charged and surface $B$ positive. Note that $\Delta F_{AB}^{+-} = (F_A^+ + F_B^-)$ - $(F_A^- + F_B^+)$ relates to the exchange $A^- + B^+ \to A^+ + B^-$, but also, $\Delta F_{AB}^{+-} = (F_A^+ - F_B^+)$ - $(F_A^- - F_B^-)$ reflects a difference in relative partitioning between surfaces of the ions. As such, contact charging can arise even if both ions favor the same surface given disparity in transfer free energies. Consequently, $\Delta F_{AB}^{+-}$ predicts the direction of charge-transfer between contacting surfaces if the charge-carrier species are $H_3O^+$ and/or $OH^-$ and populations are thermodynamically controlled and charge-transfer events are independent.

To test this hypothesis, we consider six commodity polymers (Fig. 1C): polytetrafluoroethylene (PTFE), polyethylene (PE), polyvinyl chloride (PVC), poly(methyl methacrylate) (PMMA), Nylon 66 (N66), and polyvinyl alcohol (PVA). These polymers are relevant to prior contact charging experiments[28,33,60-64], and our recent work illustrates distinct wetting behavior arising from chemically and morphologically specific water-polymer surface interactions[56]. As in ref. 56, we consider amorphous surfaces (for all six polymers), crystalline surfaces (denoted N66*, PE*, and PVC*), and surfaces featuring different tacticity (PVA† denoting isotactic PVA); calculations are performed for various droplet sizes. The combination of surface chemistry, morphology, and droplet sizes is expected to yield many distinct nanoenvironments that influence water-ion free-energies. Ultimately, $\Delta F_{AB}^{+-}$ is computed for all pairwise combinations to predict thermodynamic preference for water-ion transfer (see Methods).

The hypothesis is evaluated by comparison to experimental triboelectric series, which organize materials according to their relative propensity to acquire charges during contact charging[3]. Conventionally, triboelectric series are represented in a one-dimensional progression based on relative propensity to acquire positive/negative charge, although results do not always neatly and consistently organize in this manner. We reference three previously reported triboelectric series that feature the polymers in this study as 'S1'[3], 'S2'[63], and 'S3'[61]. These three series provide relatively consistent expectations, although there are some differences and/or omissions. In S1, the ordering, from more positive to negative, is N66, PVA, PMMA, PE, PVC, and PTFE. In S2 and S3, PVA is absent, the positions of PVC and PTFE are switched in S2, and the positions of N66 and PMMA are switched in S3. Less complete polymer triboelectric series can be formulated from elsewhere and display overall similar trends (Suppl. Fig. 1).

### Consistency of free-energy trends and contact charging

Figure 2A depicts a triboelectric matrix derived from $\Delta F_{AB}^{+-}$ values; this matrix is obtained from molecular dynamics simulations to extract the

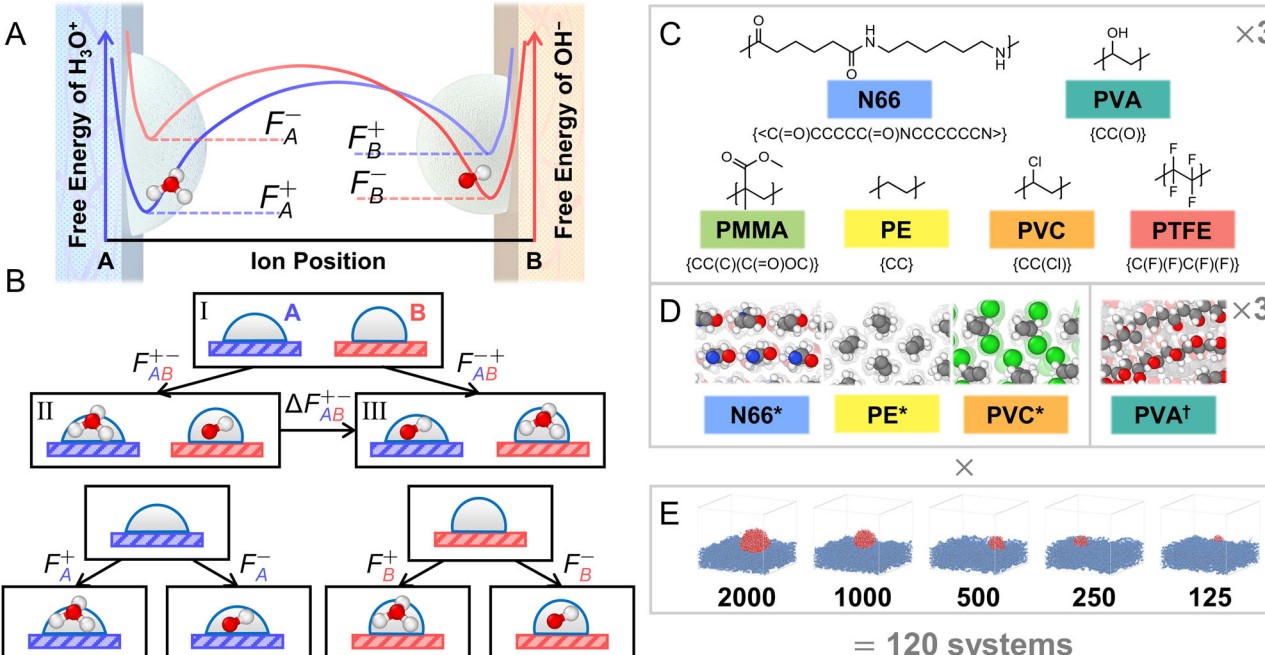

**Fig. 1 | Overview of hypothesis and systems. A** Schematic depicting how the free energy of water-ions ($H_3O^+$ and $OH^-$) may vary between two polymer surfaces. The free energy of $H_3O^+$ is represented in blue and that of $OH^-$ in red. Differences in free energy result in a thermodynamic driving force for preferential partitioning of ions between surfaces. **B** A thermodynamic framework to predict the direction of contact charging. Two model surfaces $A$ (blue) and $B$ (red) are used to illustrate the ion transfer between surfaces. The free-energy difference $\Delta F_{AB}^{+-}$ determines whether a charge-separated pair is more stable in State II with free-energy $F_{AB}^{+-}$ ($H_3O^+$ near surface $A$ and $OH^-$ near surface $B$) or State III with free energy $F_{AB}^{-+}$ ($OH^-$ near surface $A$ and $H_3O^+$ near surface $B$). **C–E** Summary of specific systems studied. The

chemical structure of the constitutional repeat unit, internal reference name, and BigSMILES string of the six polymers studied are shown (**C**). In addition to three amorphous slabs per polymer, additional crystalline slabs of N66, PE, and PVC are studied as well as three amorphous PVA slabs comprising isotactic chains; these are respectively denoted as N66*, PE*, PVC*, and PVA† (**D**). For each polymer, simulations are run using water droplets comprised of $N_w$ = 2000, 1000, 500, 250, or 125 water molecules (**E**). Molecular renderings in panel **D** and **E** are produced using OVITO[76]; carbon is gray, fluorine is blue, chlorine is green, oxygen is red, and hydrogen is white. The color-coding associated with polymer names in panel C is used throughout the text.

free energies for adding water-ions in Fig. 2B, C. To first order, the matrix is organized by material ($6 \times 6$ matrix), and results are further resolved for each $A$-$B$ into a $5 \times 5$ sub-matrix based on water-droplet size; color intensity reflects the magnitude of thermodynamic driving force. Compared to experimental triboelectric series (Suppl. Fig. 1), the simulation results broadly align with the direction of charging observed in S1, S2, and S3. In comparison to S1, simulation predictions agree with nine of fifteen material combinations, while three pairs yield inconclusive results or depend on droplet size, and three pairs exhibit opposite trends. However, when compared to S2 and S3 (which lack data for PVA), the agreement improves, as simulations predict PVC acquires negative charge over PTFE (as in S2) and N66 acquires negative charge over PMMA (as in S3). Thus, the thermodynamically informed predictions capture general trends in contact charging between polymers of different chemistry.

The few disparities between simulation predictions and empirical charging results arise in material pairings that also demonstrate experimental variability. For PVC-PTFE, S1 and S3 (and other series, see Suppl. Fig. 1) suggest that PTFE exhibits a strong tendency to acquire negative charge. However, our previous study on polymer hydrophobicity[56] indicates that water structuring and dynamics are relatively more similar between PTFE and PE than with PVC. These prior observations align with our current free-energy results, showing a vanishing $\Delta F_{AB}^{+-}$ for PE-PTFE and consistent behavior between PE-PVC and PTFE-PVC, and the experimental outcome reported via S2. Consequently, results involving PTFE may be sensitive to experimental conditions, potentially related to mechanisms not captured by simulations, such as the presence of acid and base groups post polymerization, bond breaking[26], or minor inaccuracies in molecular models. For N66-PMMA, S1 and S3 differ, with the latter aligning with

the thermodynamic predictions. Lastly, several inconsistent or inconclusive combinations involve PVA; the aqueous solubility of PVA poses an experimental challenge and is also a notable factor in our previous study[56]. Considering the substantial agreement for many material pairings and the technical challenges encountered with others, we conclude that thermodynamically driven water-ion transfer can plausibly influence polymer-polymer contact charging.

## Role of water-surface interactions

Analysis of the polymer-water interface provides nanoscale insights into the trends of water-ion free energies. To first order, we note general correlation with metrics of polymer hydrophobcity[56]. Overall, hydrogen-bonding polymers (PMMA, N66, PVA) tend to acquire positive charge more easily than non-hydrogen-bonding polymers (PE, PTFE, PVC). Furthermore, within those respective groups, increasing hydrophobicity tends to correlate with more positive-charging. To further understand these trends and how they manifest, we examine the ion-water-polymer interactions. Figure 3A compares how water, $H_3O^+$, and $OH^-$ distribute in the vicinity of chemically distinct, amorphous polymer surfaces.

Relative to $OH^-$, $H_3O^+$ tends to reside closer to the polymer-water interface, orienting its oxygen atom to maximize hydrogen-bond donation to water (Suppl. Fig. 2). Surfaces lacking hydrogen bonds, such as PE, PTFE, and PVC, allow easy access for $H_3O^+$ to the interfacial layers, explaining the similar free energy values ($F_S^+$) observed in Fig. 2B. However, $H_3O^+$ is relatively more stable (lower $F_S^+$) in proximity to hydrogen-bonding polymers (PMMA, N66, and PVA). The stronger interfacial interactions with PMMA, N66, and PVA also explain the apparent insensitivity of $F_S^+$ to droplet size (Fig. 2B), as the preferred nanoenvironment of $H_3O^+$ remains relatively consistent as droplet size

increases. Notably, $H_3O^+$ is predominantly excluded from the interfacial layer of PVA, the most hydrophilic polymer, aligning with its higher $F_S^+$ compared to PMMA and N66. This highlights an intriguing interplay between ion-polymer interactions and competing water interactions, such that ion chemical potential is not a monotonic function of hydrophilicity.

Although $OH^-$ predominantly situates in secondary interfacial layers or the bulk of water droplets, its trends also correlate with hydrophobicity and hydrogen-bonding behavior. The nearly equivalent $F_S^-$ between PE and PTFE reflects consistency in $OH^-$ distribution, which derives from their similarity in hydrophobicity and contact angles[56]. Water-ions are not notably stabilized in PE and PTFE relative to free water droplets (Suppl. Fig. 5), likely because PE and PTFE create typical hydrophobic interfaces that do not significantly impact water structure[56]. This implies that charging trends of PE and PTFE are mostly dictated by the other polymer in the contact-pair. In other words, the propensity for PTFE to acquire negative charge over N66, for example, is not due to its affinity for $OH^-$ but rather the affinity of N66 towards $H_3O^+$. Free-energy trends among N66, PVA, and PMMA align with hydrogen-bonding behavior. While N66 and PVA offer stabilizing interactions that lower $F_S^-$, PMMA only functions as a hydrogen-bond acceptor, disturbing the hydrogen-bonding network of $OH^-$ (Fig. 3C) and effectively excluding $OH^-$ from the interfacial layer of water, resulting in higher $F_S^{-56}$. In contrast to PMMA, water in proximity to PVC orients its oxygen atoms towards the surface because of the strong attraction of chlorine atoms[56], which allows water molecules to readily form hydrogen bonds with $OH^-$ in the second water layer (Fig. 3B). Thus, distinct nanoenvironments for $H_3O^+$ and $OH^-$ arise from the hydrophobicity and hydrogen-bonding behavior of the polymer surfaces, largely explaining trends in $F_S^+$ and $F_S^-$.

To further explore the sensitivity of $F_S^+$ and $F_S^-$ to interfacial interactions, we assess the role of nanoscale polymer surface morphology, which can influence hydrophobicity and hydrogen-bonding behaviors. Figure 3D, E shows the difference in $F_S^+$ and $F_S^-$ between amorphous and crystalline surfaces (for PE, PVC, and N66) and between atactic and isotactic amorphous surfaces (for PVA). Overall, the simulations capture some sensitivity of $F_S^+$ and $F_S^-$ to surface morphology, but the extent depends on polymer chemistry. The transition from PE to PE* has no notable effect, as water structuring near PE* remains similar to that of PE, resulting in nearly equivalent nanoenvironments for $H_3O^+$ and $OH^-$ and correspondingly indistinguishable free energies. However, for PVA, PVC, and N66, $F_S^+$ or $F_S^-$ can shift on scales relevant for charging predictions in Fig. 2A. Increased intra-chain hydrogen bonding and reduced hydrogen bonding with water for PVA†[56] permits more favorable water-structuring around $OH^-$, thereby increasing its stability. In N66*, the crystalline structure similarly reduces hydrogen bonding with water and results in a more hydrophobic surface, creating a less favorable nanoenvironment for $H_3O^+$ within the interfacial layer. In PVC*, enhanced chain interactions diminish interfacial water structuring, subsequently weakening interactions with $OH^-$ in secondary water layers. These findings underscore the importance of polymer-water interactions in water-ion free energies and indicate how surface heterogeneities and semicrystallinity may subtly influence water-ion transfer and contact charging.

### Connections to other charging phenomena

Although thermodynamic driving forces for ion transfer are most significant when considering different surfaces, Fig. 2B, C show that the free energy of water-ions is also influenced by droplet size, and Fig. 3D illustrates sensitivity to surface heterogeneities. The former effect is evident in the internal color variation within the diagonal material squares in Fig. 2A. Notably, for more hydrophilic polymers (PMMA, N66, and PVA), the thermodynamic driving forces are comparable to those for chemically distinct surfaces (off-diagonal squares

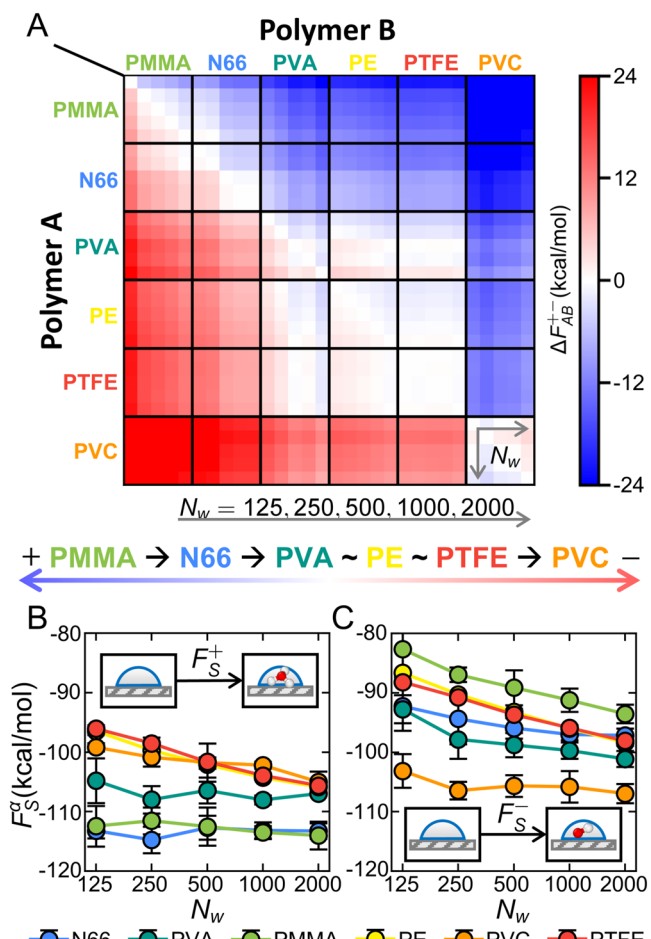

**Fig. 2 | Results of free-energy calculations for amorphous polymers. A** The thermodynamic driving force for water-ion transfer between surfaces $A$ and $B$ presented as a triboelectric matrix. The matrix is resolved $6 \times 6$ by material; each pair of materials is further resolved $5 \times 5$ accounting for differing droplet sizes. Droplet sizes ($N_w = 125, 250, 500, 1000$, and $2000$) increase left-to-right and top-to-bottom. An approximate linear triboelectric series generated from the matrix simulation is shown for reference below the matrix results. **B, C** Results of thermodynamic integration calculations to extract the free energy of adding an ion of species $\alpha$ to surface $S$ for (**B**) $H_3O^+$ and (**C**) $OH^-$. Error bars reflect statistical uncertainties reported as the standard error of the mean calculated from independent thermodynamic integration trajectories.

of Fig. 2A); Fig. 3D also conveys non-trivial differences that exceed 5 kcal/mol. These findings may have implications for contact charging of chemically identical materials[65]. If water exists on polymer surfaces as droplets of varying sizes[38] or the surfaces vary in crystallinity/patterning, these results suggest that those variabilities could create additional thermodynamic driving forces for ion redistribution and subsequent contact charging. Considering that relative humidity likely influences the distribution of droplet sizes on a surface, resulting differences in water-ion chemical potentials might account for certain humidity effects on contact charging. It is notable that the free energy of $H_3O^+$ appears less sensitive to droplet size compared to $OH^-$, particularly for hydrophilic polymers. In addition, as polymer surfaces become increasingly wet, we anticipate that any thermodynamic driving force for ion-transfer between surfaces will diminish since the contribution of the water-polymer interface will comprise an overall lesser fraction of the total ion free energy; such an effect could relate to observations of decreased contact charging at high relative humidity[40]. Although the present work does not thoroughly analyze the implications of droplet or surface heterogeneities or the precise

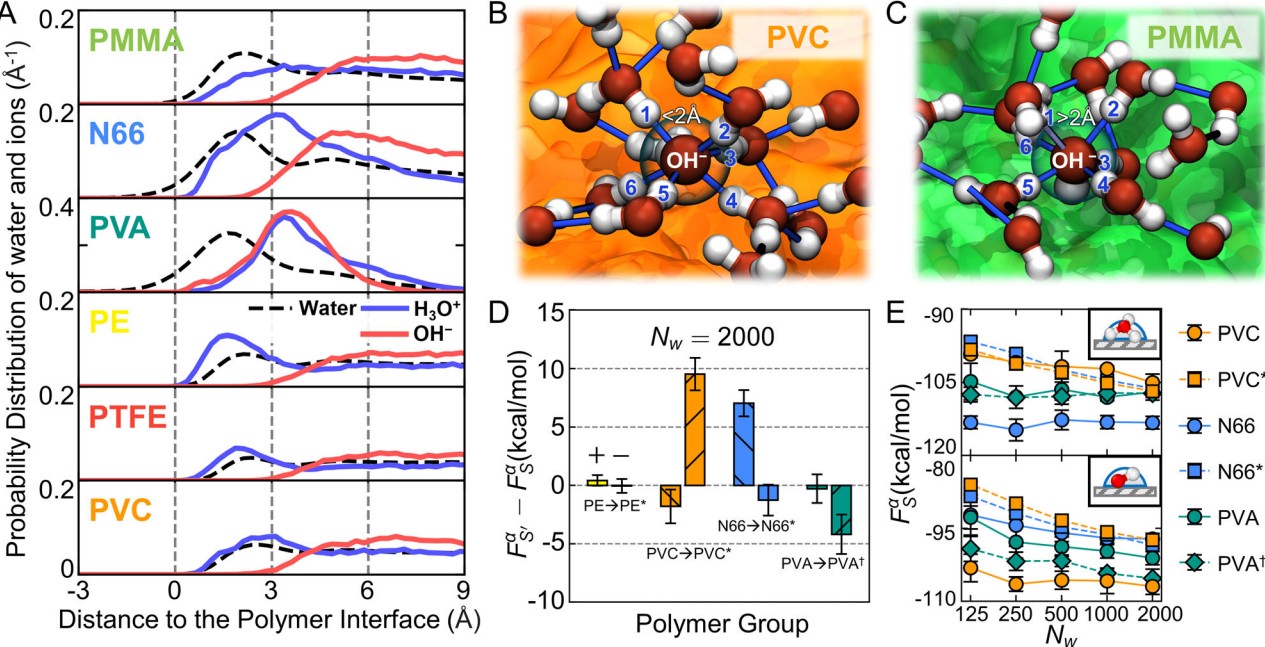

**Fig. 3 | Structural analysis of free-energy trends for $H_3O^+$ and $OH^-$ across polymers. A** Comparison of spatial distribution of water molecules, $H_3O^+$, and $OH^-$ in proximity to the polymer-water interface. The blue and red solid curves indicate the distribution of $H_3O^+$ and $OH^-$. The black dashed curve indicates the water distribution. **B**, **C** Simulation snapshots comparing $OH^-$ interactions in proximity to a (**B**) PMMA surface (orange) and (**C**) a PVC surface (green). Hydrogen-bonding interactions are indicated by thin blue bars when participating atoms are within 2 Å; interacting hydrogen atoms are labeled. The polymers are shown as orange and green surfaces in the background. Most surrounding water molecules are omitted for clarity. **D** Comparison of free energies for ion addition based on morphological changes to polymer slabs. Comparisons are made between amorphous-to-

crystalline (denoted '*') PE, PVC, and N66 and amorphous atactic-to-isotactic (denoted '†') PVA. Results are for surfaces with $N_w = 2000$ water molecules with bars grouped by material. In each group, data is organized such that $H_3O^+$ is on the left and $OH^-$ is on the right, as indicated by the '+' and '−' in the PE group. **E** Results of thermodynamic integration calculations on polymer surface morphological changes. Results between PE and PE* are statistically indistinguishable and not shown for clarity. Error bars reflect statistical uncertainties reported as the standard error of the mean calculated from independent thermodynamic integration trajectories. Molecular rendering in panels **B** and **C** used VMD[77]; oxygen is red, and hydrogen is white.

connection between droplet size and humidity, such factors could be considered in future work.

## Validation by two-surface simulations

In the preceding analysis, calculating $\Delta F_{AB}^{+-}$ involved simulating a water droplet containing a single ion above isolated polymer surfaces. As a more stringent test of these predictions, we conduct simulations with both $H_3O^+$ and $OH^-$ present between distinct polymer surfaces and assess preferential partitioning. Figure 4A illustrates the simulation setup wherein a water bridge ($N_w = 4000$) containing a $H_3O^+/OH^-$ pair forms between surfaces $A$ (top) and $B$ (bottom) separated by distance $d$. The propensity for surfaces to acquire specific charges is measured via the free energy $F_{AB}(p_z)$ where the collective variable $p_z = z_{H_3O^+} - z_{OH^-}$ is the dipole of the ionic pair in $z$-direction. As a collective variable, $p_z$ reports the relative positioning of water ions with respect to the two surfaces: more positive $p_z$ indicates $H_3O^+$ is closer to surface $A$ and $OH^-$ is closer to $B$, more negative $p_z$ indicates the opposite, and small $p_z$ suggests little to no asymmetric affinity. Similar to $\Delta F_{AB}^{+-}$, we examine the change in free energy when the dipole is flipped: $\Delta F_{AB}(p_z) \equiv F_{AB}(p_z) - F_{AB}(-p_z) = -k_B T \ln K_{AB}^{+-}(p_z)$ where $K_{AB}^{+-}$ represents a pseudo-equilibrium constant for the exchange process $A^- + B^+ \overset{K_{AB}}{\rightleftharpoons} A^+ + B^-$. Expected scenarios for $K_{AB}(p_z)$ are depicted in Fig. 4B. For example, if $K_{AB}(p_z) > 1$, $H_3O^+$ should preferentially partition towards $A$, with the expectation that $A$ becomes relatively positive and $B$ negative. The free energy $F_{AB}(p_z)$ is computed using umbrella sampling and the weighted histogram analysis method[66]; further details regarding the calculation and formulation of $K_{AB}(p_z)$ are in 'Methods.'

Results of the two-surface simulations align well with the expectations from $\Delta F_{AB}^{+-}$ (Fig. 2A) and the structural analysis (Fig. 3). Figure 4C displays $K_{AB}(p_z)$ for different pairs of materials, with row labels corresponding to surface $A$ and column labels corresponding to surface $B$. For PE-PTFE, $K_{AB}(p_z) \sim 1$, which is consistent with prior discussion on the similarity of water/ion nanoenvironments. In PVA-PTFE and PVA-PE, for which results from single-surface calculations (Fig. 2B, C) were mixed and dependent on droplet size, $K_{AB}(p_z) < 1$ indicating that $OH^-$ prefers PVA over the more hydrophobic PTFE and PE. This preference arises mainly from the recruitment of water towards the more hydrophilic surface (Suppl. Fig. 4) rather than surface-specific interactions. The remaining pairs yield $K_{AB}(p_z) > 1$, indicating enhanced thermodynamic stability of $H_3O^+$ closer to surface $A$ (row) and for $OH^-$ to be closer to $B$ (column) than the reverse situation. Thus, the two-surface simulations provide valuable validation for the overall thermodynamic framework and offer more direct support of thermodynamically driven water-ion transfer as a mechanism of contact charging.

## Discussion

Molecular dynamics simulations were used to investigate thermodynamically driven water-ion transfer as a mechanism of contact charging between insulating polymers. The ubiquity of water, correlations with hydrophobicity, and importance of humidity inform a specific hypothesis: distinct nanoenvironments for water proximate to polymer surfaces generate chemical-potential gradients that govern asymmetric transfer of water-ions upon contact (Fig. 1A, B). To investigate this hypothesis, we calculated free energies of water-ions in

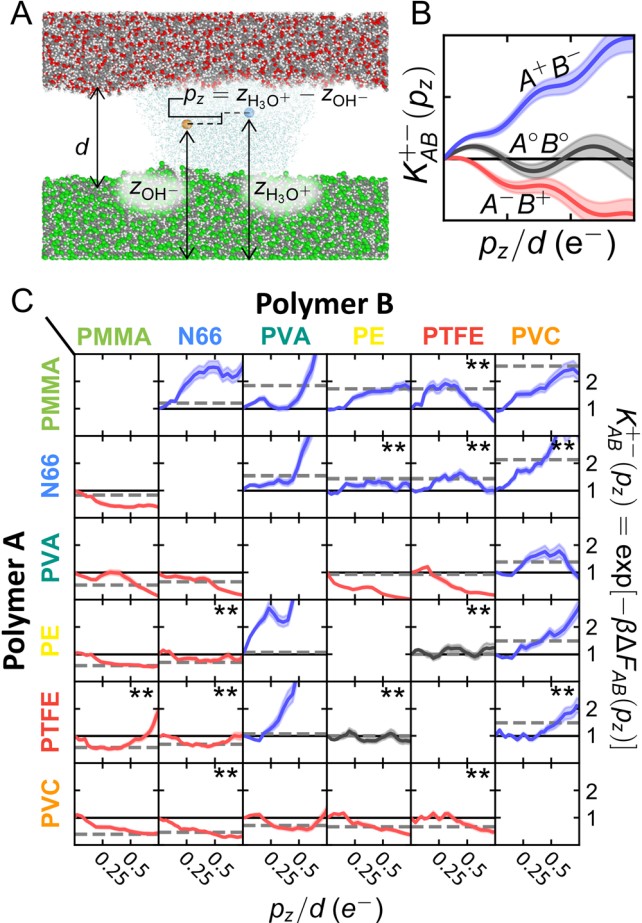

**Fig. 4 | Explicit partitioning of a H₃O⁺/OH⁻ pair between two polymer surfaces.**
**A** Simulation snapshot illustrating the general system setup for calculations. A water bridge of $N_w = 4000$ water molecules forms between two polymer slabs positioned a distance $d$ away, allowing for diffusion of $H_3O^+$ and $OH^-$ between two surfaces, $A$ and $B$. The relative positioning of $H_3O^+$ and $OH^-$ with respect to the polymer surfaces can be monitored using $p_z$, the ionic dipole in the $z$-direction. The specific system shown corresponds to PMMA (top) and PVC (bottom) positioned at $d = 25$ Å. **B** Interpretation of the exchange constant $K_{AB}^{+-}(p_z)$. If $K_{AB}^{+-}(p_z) > 1$ (blue), $H_3O^+$ exhibits more preference for A than $OH^-$ $(A^+B^-)$; if $K_{AB}^{+-}(p_z) < 1$ (red), $H_3O^+$ exhibits more preference for B than $OH^-$ $(A^-B^+)$; and if $K_{AB}^{+-}(p_z) \sim 1$ (black), there is no clear preference $(A^\circ B^\circ)$. **C** Results of $K_{AB}^{+-}(p_z)$ for different pairs of materials. Results are for $d = 25$ Å except for pairs annotated with "**", which use $d = 40$ Å to better characterize thermodynamic preference (Suppl. Fig. 3). The shaded regions reflect statistical uncertainties reported as standard error of the mean calculated from bootstrap resampling. The black solid lines indicate the position of $K_{AB}^{+-}(p_z) = 1$. The gray dashed lines show $\exp[-\beta\Delta F_{AB}^{+-}/40]$ from single-surface calculations to compare the direction of ions; the scale factor is chosen to present all data on the same scale. Molecular rendering in panel A is produced using OVITO[76]; carbon is gray, chlorine is green, oxygen is red, and hydrogen is white.

water droplets on chemically and structurally distinct polymer surfaces; these were subsequently used to predict the thermodynamically preferred direction of contact charging between various commodity polymers (Figs. 2A and 3D). Despite the simplicity of the calculations, which technically relate to the first ion-transfer event on a pristine surface and ignore kinetic factors, the predictions align well with many results of experimental triboelectric series (Suppl. Fig. 1). Subsequent simulations that directly examine partitioning of $H_3O^+$ and $OH^-$ between two surfaces offer further support (Fig. 4). The molecular resolution afforded by the simulations reveals key interactions and properties, such as surface hydrophobicity and hydrogen-bonding capabilities, that underlie relative affinities of ions to specific surfaces

(Fig. 3A–C). While other contact-charging mechanisms should not be disregarded, these results emphasize the plausibility of thermodynamic driving forces with well-defined molecular underpinnings in contact charging between insulating materials, such as polymers.

The findings offer valuable insights into the complex phenomenon of contact electrification and highlight opportunities to explore further implications across scientific and technological domains. Coupling molecular simulation with free-energy calculations can be extended to explore other aspects of contact charging, including the role of humidity[26,36,40,67], temperature[33], external electric fields[36], ion correlations, and local geometry[40,64]. Additionally, there are potential implications for contact charging between chemically identical materials, particularly regarding variations in free-energy due to differences in droplet sizes and surface morphology, though further investigation is required to ascertain their precise relevance. Moreover, future study could explore kinetic factors like asymmetric ion diffusion[50] and their interplay with thermodynamic considerations, such as ion distribution within a droplet or free-energy barriers formed during material contact. These kinetic factors could influence results based on the choice of reference probe materials and would not be explainable within a thermodynamic framework. Lastly, molecular simulations of the kind used here can provide chemically specific parameters for macroscopic models of contact charging, enabling quantitative comparisons with experiments and enhanced understanding.

## Methods
### Molecular dynamics simulations
All molecular dynamics simulations were conducted using the LAMMPS simulation package (version 3, Mar 2020)[68]. Polymers were described by parameters from the all-atom Optimized Potentials for Liquid Simulations (OPLS-AA) force field[69,70], while water was described using the extended simple point charge model (SPC/E)[71,72]. The water ions were modeled using a non-polarizable force-field designed to be used in conjunction with the SPC/E water model and parameterized to reproduce experimental solvation free energies and activities of $H_3O^+$-Cl⁻ and Na⁺-OH⁻ salt solutions[73]. Preparation of polymer-water systems includes generation of polymer surface, arrangement of water molecules on the surface (more detailed information could be found in our previous work[56]), and the addition of either $H_3O^+$ or $OH^-$ at the center-of-mass of the water droplet as required. PTFE is terminated with a trifluoromethyl group, while all other polymers are terminated with methyl groups. We note that the polymer structures are idealized in the sense that they do not reflect realistic synthetic procedures, which may result in branching structures, cross-linking, and acidic/basic terminal end groups. Simulation cells were periodic in the $x$ and $y$ directions but non-periodic in $z$; Ewald summation was accomplished via the approach of Yeh and Berkowitz[74] with extension to non-neutral systems by Ballenegger et al.[75]. After initial preparation, systems were simulated for 20 ns to generate initial configurations. Subsequently, trajectories of 40 ns were run to analyze the distribution of ions and water near polymer interfaces. More detailed information regarding simulation procedures and calculations are provided in the Supplementary Note 5.

### Single-surface free-energy calculations
The free energy associated with adding an ion of type $\alpha$ to a water droplet on surface $S$, $F_S^\alpha$, was calculated using thermodynamic integration. It was practically implemented using 12-point Gauss-Legendre quadrature for each ion, following the approach of ref. 56, which calculates the excess chemical potential of water. Simulations at each quadrature node started from the final configuration of the 20-ns equilibration trajectory. Each simulation was run for 6 ns, of which the last 5 ns were used to estimate ensemble averages. More detailed information regarding simulation procedures and calculations are provided in the Supplementary Note 6.

## Two-surface free-energy calculations

The free energy as a function of ionic dipole within a water bridge between surfaces $A$ and $B$, $F_{AB}(p_z)$, was calculated using umbrella sampling with statistical re-weighting via the weighted histogram analysis method[66]. Two-surface systems were prepared by combining two previously equilibrated polymer-water systems, mirroring the coordinates of one system across the $xy$-plane and shifting it vertically by a specified distance $d$, which was set as the average distance between polymer interfaces. Data was collected across 36 windows that each employ a harmonic biasing potential on $p_z$. The biasing potentials utilized spring constants of 47.8011 kcal/mol and equilibrium positions at −35 to 35 Å in 2 Å increments. To prevent pairing of $H_3O^+$ and $OH^-$ at small $p_z$, the force-field interaction between oxygen atoms on $H_3O^+$ and $OH^-$ was adjusted to ensure that the two ions would not bind (Suppl. Fig. 6). This modification focused analysis on ion affinity for surfaces without conflation from ionic attraction, which was not the primary focus here, and also outside the realm of application of the force-field, which does not describe recombination into neutral water species. Consequently, $F_{AB}(p_z)$ is conditional on the ions remaining separate species.

For all calculations, simulations are first run for 10 ns to equilibrate the surface-water-surface geometry. Biasing potentials were subsequently imposed for each window, and trajectories were run for 15 ns. Trajectories for windows with $|p_z| < 10$ Å were extended for an additional 7.5 ns to enhance convergence. Initially, calculations were performed at $d = 25$ Å for all surfaces. However, for some pairings (N66-PE, N66-PTFE, N66-PVC, PE-PTFE, PMMA-PTFE, and PVC-PTFE), the resulting $F_{AB}(p_z)$ was relatively flat because ions could readily interact with both surfaces. For these surfaces, additional calculations were conducted at $d = 40$ Å to better distinguish surface affinities between $H_3O^+$ and $OH^-$; calculations at greater separations yielded similar results (Suppl. Fig. 3). More detailed information regarding simulation procedures and calculations are provided in the Supplementary Note 7.

## Data availability

The process data generated in this study have been deposited in the figshare database under accession code https://doi.org/10.6084/m9.figshare.24217161.

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

## Acknowledgements

H.Z. and M.A.W. acknowledge support from a Princeton Innovation Grant via the Project X Fund. We also acknowledge support from the "Chemistry in Solution and at Interfaces" (CSI) Center funded by the U.S. Department of Energy through Award No. DE-SC0019394. Simulations and analyses were performed using resources from Princeton Research Computing at Princeton University, which is a consortium led by the Princeton Institute for Computational Science and Engineering (PICSciE) and Office of Information Technology's Research Computing.

## Author contributions

M.A.W., H.Z., and S.S. designed research; H.Z. performed research; H.Z. and M.A.W. analyzed data; M.A.W., H.Z., and S.S. discussed results; H.Z. and M.A.W. wrote the paper; M.A.W, H.Z., and S.S. edited the paper.

## Competing interests

The authors declare no competing interests.
