## [Peer Review File · Nature Communications]

Thermodynamic driving forces in contact electrification between polymeric materialsReviewer #1 (Remarks to the Author):

Contact electrification is what lies behind the simple "experiment" of rubbing a balloon on one's hair. Although it seems very simple, the science behind this is totally unknown -- for example, its hotly debated whether electrons or ions are transferred between surfaces to create the charging. Many investigations purport to show proof of either electrons or ions, but upon closer examination the evidence presented is ambiguous, and usually based on showing the fit of some ad hoc/phenomenological model.

This paper is one of the best I've ever seen in regard to addressing this question, as they use *first-principles* molecular simulations to test the hypothesis that OH⁻ and/or H₃O⁺ ions are responsible for the charging. By first-principles, I mean the simulations just set up collections of atoms and then see what results are obtained, without recourse to any assumed models. The simulations appear to be very well done, using the most appropriate simulation techniques (eg, umbrella sampling).

I recommend publication of this paper. I have a few small suggestions below.

1. I find the way the results are presented in Figs 2 and 4 is confusing. Rather than list materials according to the arbitrary list determined by M1, I suggest the authors order things according to the triboelectric series they find in their simulations (ie, from positive to negative: PMMA, nylon, PTFE/PE (tie), PVA, PVC). The figures will then look much clearer.
2. I think part D of Fig 1 should be removed. First, the rest of Fig 1 shows their simulation methodology, while part D shows experimental work of other investigators, and its confusing to have both in same figure (I initially thought these were the simulation results compared to previous triboelectric series). Furthermore, there are some things that are misleading/unnecessary about part D:
 - a. misleading: I believe M1 and M2 did NOT carry out pairwise charging experiments as indicated by this figure. Rather, they ordered materials by the sign and magnitude of charging that occurs when these materials are contacted by a single "probe material". I don't think they tested which materials charge positive and which negative when pairs of materials are contacted, which is what these plots are implying
 - b. unnecessary: Its simpler just to list the triboelectric series from each previous investigation since they are linear -- these 2d pictures don't add any new info.
3. As noted above, I believe some of the previous experimental studies ordered materials by the sign and magnitude of charging that occurs when these materials are contacted by a single "probe material"... is this ordering expected to be the same as the ordering based on which material charges positive/negative in pairwise contacts? Could there be different factors that affect magnitude of charging and direction of charging? I think this would be an interesting question to comment on in the paper.

Reviewer #2 (Remarks to the Author):

My recommendation is publication after minor revision.

The subject is of high relevance and the contributions are important for the field of contact electrification. I really enjoy reading the manuscript, which is very well written, results are clearly presented and the novelty is appropriated for Nature Communication. I also enjoyed the matrixes in red-blue which clarify the comparison between polymers. Therefore, my congratulations to the authors.

However, I ask them to discuss the following issues before accepting:

1) The authors must indicated in the Title and Introduction that the subjetc of the work is contact electrification between two polymers. One central point is that contact electrification between a polymer and an aqueous solution (water-polymer contact electrification) have been already explored in the following articles by several groups:

- Group of Prof. Helseth (Norway). Helseth, L. E. A Water Droplet-Powered Sensor Based on Charge Transfer to a Flow-through Front Surface Electrode. Nano Energy 2020, 73, 104809.

And there is a more recent paper of Helseth in Langmuir 2022 or 2023.

- Group of Prof. Butt (Germany): Stetten, A. Z.; Golovko, D. S.; Weber, S. A. L.; Butt, H.-J. Slide Electrification: Charging of Surfaces by Moving Water Drops. *Soft Matter* 2019, 15 (43), 8667–8679. <https://doi.org/10.1039/C9SM01348B>.

- Group of Prof. Negri (Argentina)
Liquid-Polymer Contact Electrification: Modelling the Dependence with pH and Added-Salt Concentration of Surface Charges and Zeta Potential. Sosa, Mariana Daniela; D'Accorso, Norma; Martínez Ricci, María; Negri, R. Martín. *Langmuir* (2022), 38, 29, 8817–8828. <https://doi.org/10.1021/acs.langmuir.2c00813>.

Sosa, M. D.; Martínez Ricci, M. L.; Missoni, L. L.; Murgida, D. H.; Cánneva, A.; D'Accorso, N. B.; Negri, R. M. Liquid–Polymer Triboelectricity: Chemical Mechanisms in the Contact Electrification Process. *Soft Matter* 2020, 16 (30), 7040–7051. <https://doi.org/10.1039/D0SM00738B>.

These articles must be cited and discussed since they analyse the origin of charge transfer between an inert polymer and a water drop. In particular, note that Sosa et al, *Langmuir* 2022, introduce an equilibrium model discussing the origin of H₃O⁺ and/or OH⁻ groups between PTFE and water. They include a mention to the presence of acid-base groups in real polymers, due to radical polymerization initiators, for instance in PTFE, which are not impurities but intrinsic components of PTFE.

Hence, i) please cite all these papers and discuss it in terms of the thermodynamics driven forces for ionic transfer and ii) pay attention when comparing with real polymers which must have terminal acid-base groups providing H₃O⁺ and OH⁻ species.

In what follows, note that all my questions are concerning with the molecular interactions between water molecules and/or ions with the functional groups of the polymers.

2) The authors must explicitly indicate which is the possible origin of the water molecules, H₃O⁺ and OH⁻ ions on the surface of non-polar polymers. Are they coming from the ambiente humidity? In that case, is the transferred charge between two polymer dependent of the relative humidity? Do the charge increase with RH or no? (a decrease with RH was indicated by the pioneer paper of Whiteside and in Sosa et al 2020 also, for the case of water-polymer contact electrification).

3) Please, discuss why the presence (adsorption?) of water molecules, H₃O⁺, OH⁻, etc decreases the surface free energy of non-polar polymers, particularly in PE and PTFE which do not have CO or Cl groups. Are those species "attracted" by the polymers? Note that it is not clear the chemical mechanism of interaction between PTFE and PE with those species. In the case of PTFE: what is the role of F atoms? Do you expect some kind of H-bond between H atoms of H₂O and the F atoms of PTFE? Or only dispersive forces are present? If only dispersive forces are present: why PTFE prefers to get negatively charged, that is PTFE prefers the OH⁻ than the H₃O⁺?

4) Related to the above: why PTFE is so strong for contact electrification? Is it possible to analyse naive questions such as why PTFE (and not PE) is at the end of the tribological series? Please, discuss this.

5) Is there any relationship-trend between the free energies, F_s and the WCA? This is a difficult topic because of the presence of carboxylic groups in real samples as mentioned by Sosa et al...but do you have simulations of WCA on PTFE, PE, etc?

6) At the Introduction, the authors mention "patches" of water. Please, indicate what do you understand or have in mind about these patches. The presence of patches suggest collective effects of water molecules which can modify surface resistance...do you agree? Please, discuss and clarify.

Reviewer #3 (Remarks to the Author):

This manuscript presents a simulation study of contact electrification in hydrated polymer surfaces. The study focuses specifically on evaluating the hypothesis that contact electrification is the result of preferential solvation of water ions (OH^- and H_3O^+) at different polymer surfaces. The manuscript presents a method for determining the direction of contact electrification and applies this method to evaluate the molecular origins of observations that have been previously reported in the literature. The method is clearly presented and appears to be effective at estimating the thermodynamic driving forces involved in contact charging via water ion transfer. Furthermore, the results are interesting and likely to appeal to a broad range of chemists. I think the manuscript is suitable for publication in Nature Communications provided that the authors address the following minor issues.

1. I would recommend that the authors reduce the length of the caption of Figure 1. Much of the explanatory text in this caption is essentially duplicated within the main text.
2. The theory neglects the free energy cost of building up a space-charge region at the interface. I would imagine that these effects may be small, but could probably be estimated from a simple model of screened electrostatics. It would be useful for the authors to attempt to quantify the Coulomb contribution to this problem.
3. The label for the scale bar in Figure 2A appears to be missing a “\Delta”
4. Perhaps it’s just me, but the snapshots in Figure 3B are difficult to interpret. Maybe the background material creates a lot of visual clutter.
5. I am curious about how the free energy differences might depend on the presence and concentration of supporting electrolyte.
6. In Figure 4, the authors present calculations of water ion pair separation between two surfaces. It would be natural to compare the free energy differences calculated in these two-surface systems to those calculated by combining the free energies of single surface systems (i.e., as presented in Figure 2). Unfortunately, the results in Figure 4 are presented in terms of equilibrium constants, which makes this side-by-side comparison difficult. Is it possible to generate a plot that makes a side-by-side comparison easier, or is there a reason that one would not expect those comparisons to coincide?
7. In the caption of Figure 4(B) there is an apparent typesetting error (upside-down question mark).

RESPONSE TO REVIEWER 1;
MANUSCRIPT ID: NCOMMS-23-46417

- **Reviewer Comments:** “Plain, quoted text”
- **Author Reply:** *Italicized font*
- **Changes to the manuscript:** Boxed text with revised text in red
All page locations reference the manuscript with markups shown.

“Contact electrification is what lies behind the simple “experiment” of rubbing a balloon on one’s hair. Although it seems very simple, the science behind this is totally unknown – for example, its hotly debated whether electrons or ions are transferred between surfaces to create the charging. Many investigations purport to show proof of either electrons or ions, but upon closer examination the evidence presented is ambiguous, and usually based on showing the fit of some ad hoc/phenomenological model.

This paper is one of the best I’ve ever seen in regard to addressing this question, as they use *first-principles* molecular simulations to test the hypothesis that OH⁻ and/or H₃O⁺ ions are responsible for the charging. By first-principles, I mean the simulations just set up collections of atoms and then see what results are obtained, without recourse to any assumed models. The simulations appear to be very well done, using the most appropriate simulation techniques (eg, umbrella sampling).

I recommend publication of this paper. I have a few small suggestions below.”

Author Reply: *We are very grateful reviewer’s positive assessment of the work and recommendation for publication. They have neatly articulated our motivations and strategy for the study, which we view as rewarding. Below, we enumerate specific changes that we have made to improve the manuscript based on the reviewer’s suggestions.*

“1. I find the way the results are presented in Figs 2 and 4 is confusing. Rather than list materials according to the arbitrary list determined by M1, I suggest the authors order things according to the triboelectric series they find in their simulations (ie, from positive to negative: PMMA, nylon, PTFE/PE (tie), PVA, PVC). The figures will then look much clearer.”

Author Reply: *We appreciate the suggestion. We have reorganized the data in Fig. 2 to comport with the triboelectric series suggested by our thermodynamic calculations. Fig. 4 is already arranged in this manner. The updated Fig. 2 is shown below.*

Figure 2: Results of free-energy calculations for amorphous polymers. (A) The thermodynamic driving force for water-ion transfer between surfaces A and B presented as a triboelectric matrix. The matrix is resolved 6×6 by material; each pair of materials is further resolved 5×5 accounting for differing droplet sizes. Droplet sizes ($N_w = 125, 250, 500, 1000,$ and 2000) increase left-to-right and top-to-bottom. An approximate linear triboelectric series generated from the matrix simulation is shown for reference below the matrix results. (B) Results of thermodynamic integration calculations to extract F_S^{α} , the free energy of adding an ion of species α to surface S . Results for adding H_3O^+ are shown at the left and OH^- are at the right. Error bars reflect statistical uncertainties reported as the standard error of the mean calculated from independent thermodynamic integration trajectories.

“2. I think part D of Fig 1 should be removed. First, the rest of Fig 1 shows their simulation methodology, while part D shows experimental work of other investigators, and its confusing to have both in same figure (I initially thought these were the simulation results compared to previous triboelectric series). Furthermore, there are some things that are misleading/unnecessary about part D: a. misleading: I believe M1 and M2 did NOT carry out pairwise charging experiments as indicated by this figure. Rather, they ordered materials by the sign and magnitude of charging that occurs when these materials are contacted by a single ”probe material”. I don’t think they tested which materials charge positive and which negative when pairs of materials are contacted, which is what these plots are implying b. unnecessary: Its simpler just to list the triboelectric series from each previous investigation since they are linear – these 2d pictures don’t add any new info.”

Author Reply: *We broadly agree with this criticism and wish to avoid any erroneous presumptions about the data or their origin. Consequently, we have removed panel D from Fig. 1, as shown below. Instead, the M1 series is now described in the text with appropriate reference to the other series as providing somewhat different rankings. In the text, we have emphasized that these series are experimental in origin. In the Supporting Information, we have also added the simple linear series. However, we have also preserved the representation of the triboelectric matrix in the SI for ease of comparison to Fig. 2. Because of the context, there should not be the same risk of conflation of simulation results/experimental data as before. Nevertheless, because the experiments may not be conducted between specific pairs of materials, we have noted this as an idealistic representation. In the framework of thermodynamics, even if the charging were completely mediated by a third “probe” material, the equilibrium condition would reflect the same charging pattern as if the target pair had been in direct contact.*

Figure 1: Overview of hypothesis and systems. (A) Schematic depicting how the free energy of water ions (H_3O^+ and OH^-) may vary between two polymer surfaces. Differences in free energy result in a thermodynamic driving force for preferential partitioning of ions between surfaces. (B) A thermodynamic framework to predict the direction of contact charging. The free-energy difference ΔF_{AB}^{+-} determines whether a charge-separated pair is more stable in State II with free-energy F_{AB}^{+-} (H_3O^+ near surface A and OH^- near surface B) or State III with free energy F_{AB}^{-+} (OH^- near surface A and H_3O^+ near surface B). (C) Summary of specific systems studied. The chemical structure of the constitutional repeat unit, internal reference name, and BigSMILES string of the six polymers studied are shown (top). In addition to three amorphous slabs per polymer, additional crystalline slabs of N66, PE, and PVC are studied as well as three amorphous PVA slabs comprising isotactic chains; these are respectively denoted as N66*, PE*, PVC*, and PVA† (middle). For each polymer, simulations are run using water droplets comprised of $N_w = 2000, 1000, 500, 250, \text{ or } 125$ water molecules (bottom). Molecular renderings in panel B are produced using OVITO (49); carbon is gray, fluorine is blue, chlorine is green, oxygen is red, and hydrogen is white. The color-coding associated with polymer names in panel C is used throughout the text.

Page 5: Conventionally, triboelectric series are represented in a one-dimensional progression based on relative propensity to acquire positive/negative charge, although results do not always neatly and consistently organize in this manner. We reference three previously reported triboelectric series that feature the polymers in this study as ‘S1’ (3), ‘S2’ (64), and ‘S3’ (62). These three series provide relatively consistent expectations, although there are some differences and/or omissions. In S1, the ordering, from more positive to negative, is N66, PVA, PMMA, PE, PVC, PTFE. In S2 and S3, PVA is absent, the positions of PVC and PTFE are switched in S2, and the positions of N66 and PMMA are switched in S3. Less complete polymer triboelectric series can be formulated from elsewhere and display overall similar trends (see SI Appendix Fig. S1).

Page 6: To first order, the matrix is organized by material (6×6 matrix), and results are further resolved for each A - B into a 5×5 sub-matrix based on water-droplet size; color intensity reflects the magnitude of thermodynamic driving force. Compared to experimental triboelectric series (SI Appendix, Fig. S1), the simulation results broadly align with the direction of charging observed in S1, S2, and S3.

“3. As noted above, I believe some of the previous experimental studies ordered materials by the sign and magnitude of charging that occurs when these materials are contacted by a single “probe material”... is this ordering expected to be the same as the ordering based on which material charges positive/negative in pairwise contacts? Could there be different factors that affect magnitude of charging and direction of charging? I think this would be an interesting question to comment on in the paper.”

Author Reply: *These are interesting questions. As expressed in response to the second comment, if each material and the probe material are considered to be in equilibrium, then there should be no difference between the ordering expressed from the experiment involving a probe material versus that involving the two materials explicitly. Nevertheless, if the experiments are affected by kinetic factors, which is likely to some extent, then differences could indeed arise and depend on the probe material. We believe this could motivate some future experiments. While the current work emphasizes the plausibility of charging with thermodynamic arguments, we plan to address some kinetic factors in future studies. We have added a couple statements to the text regarding these points.*

Additionally, our study technically examines the thermodynamic driving force for the transfer of the first charge or for charge-transfer events that are approximately independent. In experiments, the effect of the electric field that develops as more and more charges are transferred becomes important, but we do not consider the effect of prevailing electric field. We now note this limitation and possibility for future study in the revised manuscript.

Despite the simplicity of the calculations, which technically relate to the first ion-transfer event on a pristine surface and ignore kinetic factors, the predictions align remarkably well with many results of experimental triboelectric series (Fig. S1). Subsequent simulations that directly examine partitioning of H_3O^+ and OH^- between two surfaces offer further support (Fig. 4).

Page 11: Coupling molecular simulation with free-energy calculations can be extended to explore other aspects of contact charging, including the role of humidity (26, 36, 40, 68), temperature (33), external electric fields, (36) ion correlations, and local geometry (40, 65). Additionally, there are potential implications for contact charging between chemically identical materials, particularly regarding variations in free-energy due to differences in droplet sizes and surface morphology, though further investigation is required to ascertain their precise relevance. Moreover, future study could explore kinetic factors like asymmetric ion diffusion (51) and their interplay with thermodynamic considerations, such as ion distribution within a droplet or free-energy barriers formed during material contact. These kinetic factors could influence results based on the choice of reference probe materials and would not be explainable within a thermodynamic framework.

RESPONSE TO REVIEWER 2;
MANUSCRIPT ID: NCOMMS-23-46417

- **Reviewer Comments:** “Plain, quoted text”
- **Author Reply:** *Italicized font*
- **Changes to the manuscript:** Boxed text with revised text in red
All page locations reference the manuscript with markups shown.

“My recommendation is publication after minor revision. The subject is of high relevance and the contributions are important for the field of contact electrification. I really enjoy reading the manuscript, which is very well written, results are clearly presented and the novelty is appropriated for Nature Communication. I also enjoyed the matrixes in red-blue which clarify the comparison between polymers. Therefoire, my congratulations to the authors. However, I ask them to discuss the following issues before accepting:”

Author Reply: *We appreciate the reviewer’s comments and recommendation for publication. Below, we enumerate specific changes that we have made to improve the manuscript based on the reviewer’s suggestions.*

“1) The authors must indicated in the Title and Introduction that the subject of the work is contact electrification between two polymers. One central point is that contact electrification between a polymer and an aqueous solution (water-polymer contact electrification) have been already explored in the following articles by several groups: - Group of Prof. Helseth (Norway). Helseth, L. E. A Water Droplet-Powered Sensor Based on Charge Transfer to a Flow-through Front Surface Electrode. *Nano Energy* 2020, 73, 104809. And there is a more recent paper of Helseth in *Langmuir* 2022 or 2023.

- Group of Prof. Butt (Germany): Stetten, A. Z.; Golovko, D. S.; Weber, S. A. L.; Butt, H.-J. Slide Electrification: Charging of Surfaces by Moving Water Drops. *Soft Matter* 2019, 15 (43), 8667–8679. <https://doi.org/10.1039/C9SM01348B>.

- Group of Prof. Negri (Argentina) Liquid-Polymer Contact Electrification: Modelling the Dependence with pH and Added-Salt Concentration of Surface Charges and Zeta Potential. Sosa, Mariana Daniela; D’Accorso, Norma; Martínez Ricci, María; Negri, R. Martín. *Langmuir* (2022), 38, 29, 8817–8828. <https://doi.org/10.1021/acs.langmuir.2c00813>.

Sosa, M. D.; Martínez Ricci, M. L.; Missoni, L. L.; Murgida, D. H.; Cánneva, A.; D’Accorso, N. B.; Negri, R. M. Liquid–Polymer Triboelectricity: Chemical Mechanisms in the Contact Electrification Process. *Soft Matter* 2020, 16 (30), 7040–7051. <https://doi.org/10.1039/D0SM00738B>.

These articles must be cited and discussed since they analyse the origin of charge transfer between an inert polymer and a water drop. In particular, note that Sosa et al , *Langmuir* 2022, introduce an equilibrium model discussing the origin of H₃O⁺ and/or OH⁻ groups between PTFE and water. They include a mention to the presence of acid-base groups in real polymers, due to radical polymerization initiators, for instance in PTFE, which are not impurities but intrinsic components of PTFE. Hence, i) please cite all these papers and discuss it in terms of the thermodynamics driven forces for ionic transfer and ii) pay attention when comparing with real polymers which must have terminal acid-base groups providing H₃O⁺ and OH⁻ species.”

Author Reply: *We understand the perspective of the reviewer. Of course, we agree that contact electrification is an important process that pervades all manner of materials and phases. We have generally focused our discussion and referencing around a specific type of materials and manner of charging. Therefore, much of this interesting literature was neither cited nor discussed in the original manuscript. The comments regarding work by Sosa et al. (and also later points by the reviewer about real systems) do highlight a need for clarification regarding the scope and interpretation of our study. We have thoughtfully addressed the reviewer’s comments here, but we note the need to maintain a balance of retaining the specificity of our study while acknowledging some important but more peripheral work/phenomena.*

In the title, we have changed the word ‘of’ to ‘between’ to emphasize the phenomena we specifically address.

In the introduction, we have cited the works suggested by the reviewer, predominantly in the context of highlighting important water/ion effects and their origin; the mechanisms of liquid-polymer charging are beyond the scope of what we would like to introduce to appreciate the present work. Importantly, we have also added some contextualizing and clarifying statements about the manifestation of polymers in our study, presenting nearly ideal surfaces, that lack the presence of acidic/basic terminal end groups that may be featured in real systems. We respectfully submit that capturing this phenomena is beyond the scope of the present manuscript (and technically challenging from a modeling perspective at this stage). Nevertheless, we note that the presence of acid-base groups on the surface, unless in very high proportion to the balance of presented surface chemistry, is unlikely to influence our thermodynamic calculations because they are agnostic to the precise source of the ions.

*Title: Thermodynamic driving forces in contact electrification **between** polymeric materials*

*Page 2: **Mosaic patterns of charge on polymer surfaces following contact have been attributed to the presence of water patches (39), as water has been observed to only partially cover surfaces, forming patches or islands (37, 38). Effects of relative humidity on electrostatic charging highlight a potential role of water and its ions (28, 30, 37, 40) as do numerous studies related charging phenomena directly at liquid-solid interfaces (41–45).***

*Page 6: **Consequently, results involving PTFE may be sensitive to experimental conditions, potentially related to mechanisms not captured by simulations, such as the presence of acid and base groups post polymerization, bond breaking (? , 26) or minor inaccuracies in molecular models.***

*Page 13: **PTFE is terminated with a trifluoromethyl group, while all other polymers are terminated with methyl groups. We note that the polymer structures are idealized in the sense that they do not reflect realistic synthetic procedures, which may result in branching structures, cross-linking, and acidic/basic terminal end groups.***

“In what follows, note that all my questions are concerning with the molecular interactions between water molecules and/or ions with the functional groups of the polymers.”

“2) The authors must explicitly indicate which is the possible origin of the water molecules, H_3O^+ and OH^- ions on the surface of non-polar polymers. Are they coming from the ambient humidity? In that case, is the transferred charge between two polymer dependent of the relative humidity? Do the charge increase with RH or no? (a decrease with RH was indicated by the pioneer paper of Whiteside and in Sosa et al. 2020 also, for the case of water-polymer contact electrification).”

Author Reply: *With respect to the water molecules, there is evidence that water often persists on surfaces even under low-humidity conditions; this has been cited and noted in the introduction. We have constructed our study with the idea that the water droplets arise just from ambient conditions. One can anticipate that the number of water molecules in a droplet and the distribution of droplet sizes is related to the humidity. Furthermore, this relationship likely depends on the polymer surface chemistry. However, we did not specifically compute this relationship. We now note that making this connection is of interest.*

With respect to the water-ions, we again speculate these to arise from the ambient water itself. As alluded in the response to the prior query, however, the thermodynamic integration calculations are technically agnostic to the origin of the ions. Our calculations reveal what the relative driving forces would be supposing that such ions are generated. We have made a clarifying statement in this direction. Our calculations can hopefully be coupled to other theories, such as those of Sosa et al., to provide a more robust conceptual framework.

With respect to the observation regarding the decrease in charging with increase in RH, there are likely many factors. Limiting discussion to our calculations, we can speculate that increasing humidity increases the amount of water on the respective polymer surfaces and increases droplet size on average. In the limit of large water droplets or complete wetting, the free energy difference for ions to move from one surface to another will decrease because there is a lesser proportion of the water environment that is sensitive to polymer surface chemistry – it becomes dominated by the bulk-like water environment. Therefore, within a thermodynamic framework, there is less propensity for charging by this mechanism. This is a speculative point that we now briefly proffer when noting the effects of water-droplet size.

We do note that the presence of H_3O^+ and OH^- ions would be rather unlikely in dry air. The presence of water vapor in the gas allows the formation of water ion clusters (i.e., H_3O^+ and OH^- surrounded by several water molecules), which may dramatically increase the stability of the water ions. Hence, we speculate that the likelihood of the presence of water-ion clusters would increase with humidity. A study of water ion clusters in humid air through simulations would be interesting, and we will probe that in future studies.

Page 5: We suppose that the water ions arise from ambient water, as suggested by previous experimental studies (59, 60), but all calculations are agnostic to their their precise origin.

Page 5: Consequently, we suppose ΔF_{AB}^{+-} predicts the direction of charge-transfer between contacting surfaces if the charge-carrier species are H_3O^+ and/or OH^- and populations are thermodynamically controlled and charge-transfer events are independent.

Page 10: In addition, as polymer surfaces become increasingly wet, we anticipate that any thermodynamic driving force for ion-transfer between surfaces will diminish since the contribution of the water-polymer interface will comprise any overall lesser fraction of the total ion free energy; such an effect could relate to observations of decreased contact charging at high relative humidity. (40) Although the present work does not thoroughly analyze the implications of droplet or surface heterogeneities or the precise connection between droplet size and humidity, such factors could be considered in future work.

“3) Please, discuss why the presence (adsorption ?) of water molecules, H_3O^+ , OH^- , etc decreases the surface free energy of non-polar polymers, particularly in PE and PTFE which do not have CO or Cl groups. Are those species ”attracted” by the polymers? Note that it is not clear the chemical mechanism of interaction between PTFE and PE with those species. In the case of PTFE: what is the role of F atoms? Do you expect some kind of H-bond between H atoms of H_2O and the F atoms of PTFE ? Or only dispersive forces are present ? If only dispersive forces are present: why PTFE prefers to get negatively charged, that is PTFE prefers the OH^- than the H_3O^+ ?”

Author Reply: *We should clarify a couple of facets of our predictions. Water droplets indeed possess lower free-energy than free droplets in vacuum due to dispersion forces. In our previous study (10.1021/acs.jpcc.3c00616), we calculated the free-energy required to dewet droplets from various polymer surfaces. These calculations revealed similar but non-zero work was required for PE and PTFE, but it was less than all the other polymers. Induced water-orientation was slightly different between PE and PTFE in proximity to the polymer-water interface due to the disparity in size between fluorine and hydrogen; the larger fluorine weakens interaction with water in the simulations.*

More specific to the reviewer’s question, we do not believe the apparent negative charging of PTFE/PE is actually driven based on enhanced affinity to OH^- but rather by diminished affinity to H_3O^+ . Importantly, the direction of charging depends not just on affinity of a specific surface but the relative affinity. As discussed towards the end of the first paragraph on Page 5, this can be written as $\Delta F_{AB}^{+-} = (F_A^+ - F_B^+) - (F_A^- - F_B^-)$. We observe from Figure 2 that the chemical potentials of H_3O^+ on PE and PTFE are generally more positive than the other polymers. Therefore, we suppose that the first term $(F_A^+ - F_B^+)$ is most responsible for the charging trends here, while $(F_A^- - F_B^-)$ has a more varied role. We have added an explicit comment about this effect in the manuscript.

Inspired by this question, we have also performed additional calculations to extract the chemical potential of the ion in a free water droplet. These calculations reveal that the chemical potentials in the free droplet are statistically indistinguishable from those within droplets on PE or PTFE. This bolsters the idea that the free-energy trends are primarily induced by the chemistry of the other material in the contact-pair. These calculations have been added to the SI. We emphasize again, however, that these calculations are treating idealized surfaces.

Page 8: Water-ions are not notably stabilized in PE and PTFE relative to free water droplets (SI Appendix, Fig. S5), likely because PE and PTFE create typical hydrophobic interfaces that do not significantly impact water structure (57). This implies that charging trends of PE and PTFE are mostly dictated by the other polymer in the contact-pair. In other words, the propensity for PTFE to acquire negative charge over N66, for example, is not due to its affinity for OH^- but rather the affinity of N66 towards H_3O^+ .

Page S4: To further understand our single free-energy results, we performed calculations involving a free water droplet. The simulation systems are generated by placing a water molecules in a spherical geometry at the center of simulation cell; subsequent simulation procedures follow those of the single-surface free-energy calculations. Fig. S5 shows that the free energy of adding an ion to a free water droplet are statistically indistinguishable from those of adding to droplets on PE and PTFE. This implies that the ions within free water droplets are stabilized to a similar extent as on hydrophobic surfaces. By extension, these results suggest that the free-energy trends of PE and PTFE are primarily induced by chemically specific effects moreso related to the other polymer surface in the contact-pair.

“4) Related to the above: why PTFE is so strong for contact electrification ? Is it possible to analyse naif questions such as why PTFE (and not PE) is at the end of the tribological series? Please, discuss this.”

Author Reply: *This point should be clarified in the response to point #3: the hydrophobic nature of PTFE makes it exhibit lesser affinity for H_3O^+ relative to other surfaces. Of course, these calculations do not preclude other factors (kinetics or non-ideal surface terminations) from impacting the charging of PTFE, but we are not in a position to analyze that in the present work. In addition, our calculations consistently rank PVC as more likely to acquire negative charge than PTFE—so, the simulation data does not necessarily support the idea that PTFE is overwhelmingly strong in acquiring negative charge, at least for the ideal surfaces represented in our work. This is not necessarily at odds with experiment. While PTFE is at the end of the tribological series for the series presented as ‘S1’, this is not the case for ‘S2’ or other series like in Fig. S1, panel K.*

“5) Is there any relationship-trend between the free energies, Fs and the WCA ? This is a difficult topic because of the presence of carboxylic groups in real samples as mentioned by Sosa et al...but do you have simulations of WCA on PTFE, PE,etc ?”

Author Reply: *We do believe there is at least an indirect relationship between the ion free energies and water contact angle. In a previous study (Ref. 57), we have characterized different metrics of polymer hydrophobicity, including the contact angle, for the same polymers as studied here. The connection is that the physics that dictate contact angle have very relevant implications for either direct interactions with ions or indirect via effects on water structuring. We have now made these observations more explicit in the main text. As mentioned in other responses, however, these calculations are still for pristine surfaces that do not feature any kind of acid-base chemistry as a result of termination.*

Page 7: Analysis of the polymer-water interface provides nanoscale insights into the trends of water-ion free energies. To first order, we note general correlation with metrics of polymer hydrophobicity (57). Overall, hydrogen-bonding polymers (PMMA, N66, PVA) tend to acquire positive charge more easily than non-hydrogen-bonding polymers (PE, PTFE, PVC). Furthermore, within those respective groups, increasing hydrophobicity tends to correlate with more positive-charging. To further understand these trends and how they manifest, we examine the ion-water-polymer interactions.

“6) At the Introduction, the authors mention “patches” of water. Please, indicate what do you understand or have in mind about these patches. The presence of patches suggest collective effects of water molecules which can modify surface resistance ...do you agree? Please, discuss and clarify.”

Author Reply: *Our reference to patches mostly derives from claims from literature work, such as in Fig. 2 in <https://doi.org/10.1016/j.apsusc.2012.07.076> or in <https://www.science.org/doi/10.1126/science.1192907>. The major idea is that there is only partial surface coverage of water on surfaces in most ambient conditions. Because of attractive forces between water molecules, they are likely to form clusters/patches/islands rather than persist as independent adsorbed species. We believe it is indeed sensible that the nature of these patches and their distribution could influence surface resistance. Disparity in patch distribution is an element of the model proposed by Jaeger and coworkers. We have made a clarifying statement regarding the nature of the “patches” identified in the literature.*

Page 2: Mosaic patterns of charge on polymer surfaces following contact have been attributed to the presence of water patches (39), as water has been observed to only partially cover surfaces, forming patches or islands (37, 38). Effects of relative humidity on electrostatic charging highlight a potential role of water and its ions (28, 30, 37, 40) as do numerous studies related charging phenomena directly at liquid-solid interfaces (41–45).

RESPONSE TO REVIEWER 3;
MANUSCRIPT ID: NCOMMS-23-46417

- **Reviewer Comments:** “Plain, quoted text”
- **Author Reply:** *Italicized font*
- **Changes to the manuscript:** Boxed text with revised text in red
All page locations reference the manuscript with markups shown.

“This manuscript presents a simulation study of contact electrification in hydrated polymer surfaces. The study focuses specifically on evaluating the hypothesis that contact electrification is the result of preferential solvation of water ions (OH⁻ and H₃O⁺) at different polymer surfaces. The manuscript presents a method for determining the direction of contact electrification and applies this method to evaluate the molecular origins of observations that have been previously reported in the literature. The method is clearly presented and appears to be effective at estimating the thermodynamic driving forces involved in contact charging via water ion transfer. Furthermore, the results are interesting and likely to appeal to a broad range of chemists. I think the manuscript is suitable for publication in Nature Communications provided that the authors address the following minor issues.”

Author Reply: *We thank the reviewer for their recommendation for publication. We believe we have satisfactorily addressed the minor issues raised. Below, we enumerate the specific changes based on the reviewer’s suggestions.*

“1. I would recommend that the authors reduce the length of the caption of Figure 1. Much of the explanatory text in this caption is essentially duplicated within the main text.”

Author Reply: *We have streamlined the caption as requested but have preserved details to emphasize aspects of the study, in case there are readers that provide more cursory examination of the main text itself. Based on a recommendation from another referee, we have also eliminated panel D, which should help.*

Figure 1: Overview of hypothesis and systems. (A) Schematic depicting how the free energy of water ions (H_3O^+ and OH^-) may vary between two polymer surfaces. Differences in free energy result in a thermodynamic driving force for preferential partitioning of ions between surfaces. (B) A thermodynamic framework to predict the direction of contact charging. The free-energy difference ΔF_{AB}^{+-} determines whether a charge-separated pair is more stable in State II with free-energy F_{AB}^{+-} (H_3O^+ near surface A and OH^- near surface B) or State III with free energy F_{AB}^{+} (OH^- near surface A and H_3O^+ near surface B). (C) Summary of specific systems studied. The chemical structure of the constitutional repeat unit, internal reference name, and BigSMILES string of the six polymers studied are shown (top). In addition to three amorphous slabs per polymer, additional crystalline slabs of N66, PE, and PVC are studied as well as three amorphous PVA slabs comprising isotactic chains; these are respectively denoted as N66*, PE*, PVC*, and PVA[†] (middle). For each polymer, simulations are run using water droplets comprised of $N_w = 2000, 1000, 500, 250, \text{ or } 125$ water molecules (bottom). Molecular renderings in panel B are produced using OVITO (49); carbon is gray, fluorine is blue, chlorine is green, oxygen is red, and hydrogen is white. The color-coding associated with polymer names in panel C is used throughout the text.

“2. The theory neglects the free energy cost of building up a space-charge region at the interface. I would imagine that these effects may be small, but could probably be estimated from a simple model of screened electrostatics. It would be useful for the authors to attempt to quantify the Coulomb contribution to this problem.”

Author Reply: *Indeed, we do not address the aspect of a space-charge region or the development of an electric field. Our calculations report on the thermodynamic driving force for the transfer of the first ion or apply in scenarios where transfers can be treated as roughly independent. The current results alone establish the possible relevance of thermodynamic driving forces, and so explicitly reconciling the issue raised by the reviewer, which would require significant additional simulation, is somewhat outside the scope of the present work. It is likely that this effect will have a role in quantitatively predicting the magnitude of charging, but not the direction. We have added some clarifying comments to the text to highlight the issue raised, and we hope to quantitatively address the problem with additional theory and simulations in the future.*

Despite the simplicity of the calculations, which technically relate to the first ion-transfer event on a pristine surface and ignore kinetic factors, the predictions align remarkably well with many results of experimental triboelectric series (Fig. S1). Subsequent simulations that directly examine partitioning of H_3O^+ and OH^- between two surfaces offer further support (Fig. 4).

Page 11: Coupling molecular simulation with free-energy calculations can be extended to explore other aspects of contact charging, including the role of humidity (26, 36, 40, 68), temperature (33), external electric fields, (36) ion correlations, and local geometry (40, 65). Additionally, there are potential implications for contact charging between chemically identical materials, particularly regarding variations in free-energy due to differences in droplet sizes and surface morphology, though further investigation is required to ascertain their precise relevance. Moreover, future study could explore kinetic factors like asymmetric ion diffusion (51) and their interplay with thermodynamic considerations, such as ion distribution within a droplet or free-energy barriers formed during material contact. These kinetic factors could influence results based on the choice of reference probe materials and would not be explainable within a thermodynamic framework.

“3. The label for the scale bar in Figure 2A appears to be missing a Delta”

Author Reply: *Thank you. This is now corrected.*

“4. Perhaps it’s just me, but the snapshots in Figure 3B are difficult to interpret. Maybe the background material creates a lot of visual clutter. ”

Author Reply: *We have tried to improve the visual accessibility of Figure 3B and have added some text to highlight what we find may be salient features. Ultimately, we anticipate that some readers may find the snapshots useful, while others may find them less compelling.*

Page 11: While N66 and PVA offer stabilizing interactions that lower F_S^- , PMMA only functions as a hydrogen-bond acceptor, **disturbing the hydrogen-bonding network of OH^- (Fig. 3b) and** effectively excluding OH^- from the interfacial layer of water, resulting in higher F_S^- (57).

“5. I am curious about how the free energy differences might depend on the presence and concentration of supporting electrolyte. ”

Author Reply: *This is an interesting question, but similar to the issue of the space-charge, this is not ultimately in the domain of the present work. We would speculate that this is at most a second-order effect because the direct interaction with the water-ions would be agnostic to polymer chemistry. Rather, sensitivity to polymer chemistry would come by way of modifications to the water structuring in the vicinity of the polymer-water interface. Without explicit simulations, these comments are simply speculation.*

“6. In Figure 4, the authors present calculations of water ion pair separation between two surfaces. It would be natural to compare the free energy differences calculated in these two-surface systems to those calculated by combining the free energies of single surface systems (i.e., as presented in Figure 2). Unfortunately, the results in Figure 4 are presented in terms of equilibrium constants, which makes this side-by-side comparison difficult. Is it possible to generate a plot that makes a side-by-side comparison easier, or is there a reason that one would not expect those comparisons to coincide?”

Author Reply: *We agree that it is natural to want to compare these two quantities. A direct one-to-comparison is obfuscated by several factors, mostly related to the ion-ion interactions but also facets of the water channel. Nevertheless, we think it makes sense to reinforce the idea of trend consistency. We have therefore modified Fig. 4 to illustrate the results of Fig. 2 but in the vein of a partitioning coefficient. To show results on similar scale, however, it is necessary to scale the free-energy values from the single-surface calculations.*

“7. In the caption of Figure 4(B) there is an apparent typesetting error (upside-down question mark).” **Author Reply:** *Thank you. This is corrected.*

Reviewer #1 (Remarks to the Author):

The revised manuscript has effectively addressed the questions raised in my initial review

I was asked to see if the revised manuscript also effectively addressed the questions of Reviewer 3. I have reviewed the author responses to the comments of Reviewer 3, and I feel that the authors have effectively addressed these questions.

Reviewer #2 (Remarks to the Author):

The authors answered satisfactorily all my questions. My congratulations to them. Thus, in my opinion the article must be published as it is in the revised version.